# COVID-19 and Influenza Coinfection Outcomes among Hospitalized Patients in the United States: A Propensity Matched Analysis of National Inpatient Sample

**DOI:** 10.3390/vaccines10122159

**Published:** 2022-12-15

**Authors:** Ishan Garg, Karthik Gangu, Hina Shuja, Alireza Agahi, Harsh Sharma, Aniesh Bobba, Adeel Nasrullah, Prabal Chourasia, Suman Pal, Abu Baker Sheikh, Rahul Shekhar

**Affiliations:** 1Department of Internal Medicine, University of New Mexico Health Sciences Center, Albuquerque, NM 87106, USA; 2Division of Cardiology, University of New Mexico Health Sciences Center, Albuquerque, NM 87106, USA; 3Department of Medicine, Karachi Medical and Dental College, Karachi 74700, Pakistan; 4Department of Pathology, Mayo Clinic, Rochester, MN 55905, USA; 5Department of Medicine, John H. Stronger Hospital, Chicago, IL 60612, USA; 6Division of Pulmonology and Critical Care, Allegheny Health Network, Pittsburg, PA 15212, USA; 7Department of Medicine, Mary Washington Hospital, Fredericksburg, VA 22401, USA

**Keywords:** COVID-19, SARS CoV-2, influenza, N.I.S. data, coinfection, outcome

## Abstract

This study aims to provide comparative data on clinical features and in-hospital outcomes among U.S. adults admitted to the hospital with COVID-19 and influenza infection using a nationwide inpatient sample (N.I.S.) data 2020. Data were collected on patient characteristics and in-hospital outcomes, including patient’s age, race, sex, insurance status, median income, length of stay, mortality, hospitalization cost, comorbidities, mechanical ventilation, and vasopressor support. Additional analysis was performed using propensity matching. In propensity-matched cohort analysis, influenza-positive (and COVID-positive) patients had higher mean hospitalization cost (USD 129,742 vs. USD 68,878, *p* = 0.04) and total length of stay (9.9 days vs. 8.2 days, *p* = 0.01), higher odds of needing mechanical ventilation (OR 2.01, 95% CI 1.19–3.39), and higher in-hospital mortality (OR 2.09, 95% CI 1.03–4.24) relative to the COVID-positive and influenza-negative cohort. In conclusion, COVID-positive and influenza-negative patients had lower hospital charges, shorter hospital stays, and overall lower mortality, thereby supporting the use of the influenza vaccine in COVID-positive patients.

## 1. Introduction

The coronavirus disease 2019 (COVID-19) pandemic has devastated human life, healthcare systems, and the world economy. It is caused by severe acute respiratory syndrome coronavirus 2 (SARS-CoV-2). As of October 2022, it has claimed the lives of 6.5 million people and infected over 623 million cases worldwide [1]. Older adults and those with comorbidities are at a greater risk of developing severe symptoms related to COVID-19 with worse outcomes.

SARS-CoV-2 and influenza viruses are transmitted through similar mechanisms: direct contact (human-to-human transmission) and respiratory or airborne droplets [2,3,4,5,6,7,8]. To make matters worse, both diseases can present with overlapping clinical features ranging from mild flu-like symptoms, including cough, sore throat, headache, muscle aches, breathlessness, and fever, to pneumonia, acute respiratory distress syndrome (ARDS), and death [9,10,11]. Therefore COVID-19 coinfection with other lower respiratory tract pathogens can complicate management and overall prognosis in these patients. One meta-analysis on synchronous COVID-19 and Influenza infection in over 3000 patients reported a prevalence of influenza infection was 0.8% in patients with confirmed COVID-19 [12]. However, poorer outcomes have been reported with COVID-19 and influenza coinfections [13,14]. However, these comparisons have so far relied on data obtained by pooling multiple studies using disparate methodologies [13,15,16,17,18,19,20,21].

In this study, we aim to utilize the United States National Inpatient Sample (N.I.S.) database from 2020 to provide more robust comparative data on clinical features and outcomes among U.S. adults admitted to the hospitalist with COVID-19 and influenza infection as well as discuss the reasons for vaccine hesitancy and its potential impact on healthcare policies.

## 2. Materials and Methods

This was a retrospective study conducted using the N.I.S. data from 2020. Established by the agency for healthcare research and quality (AHRQ), the N.I.S. is the largest all-payer healthcare database in the United States [22]. International classification of diseases 10th—clinical modification (ICD-10-CM) codes were used to retrieve patient samples with comorbid conditions, and ICD-10 procedure codes were used to identify inpatient procedures [23]. All patients who were 18 years of age and older and admitted to the hospital with COVID-19 infection were included in this study.

The N.I.S. database contains data regarding in-hospital outcomes, procedures, and other discharge-related information. Variables were divided into patient-related, hospital-related, and indicators of illness severity as below:Patient: age, race (White, Black, Hispanic, Native American, Asian, other), sex, insurance status (Medicare, Medicaid, private insurance, self-payment, no charge), median income based on patient’s zip code, and disposition.Hospital: location, teaching status, bed size, and region.Illness severity: length of stay (L.O.S.), mortality, hospitalization cost, comorbidities, mechanical ventilation, circulatory support, and vasopressor use.

The primary outcome was in-hospital mortality. Secondary outcomes included:intubation and mechanical ventilationvasopressor uselength of stay, the financial burden on healthcare, and resource utilization.

### Statistical Analysis

Descriptive statistics were used to summarize the continuous and categorical variables. Continuous variables were summarized as mean±SD (standard deviation); categorical data as numbers and percentages. Univariate analyses for between-group comparisons used the Rao-Scott Chi-square test for categorical variables (e.g., sex and risk factors) and weighted simple linear regression for continuous variables (e.g., age). On the unmatched sample, univariate regression was used to identify independent variables (*p* ≤ 0.2), which were utilized to build a multivariate regression model. As our control group (COVID-positive but Flu-negative) had a significantly higher sample than the test group (COVID-positive and Flu-positive), we conducted a secondary analysis with propensity matching to confirm results obtained by traditional multivariate analysis. Baseline demographics (Age, race, sex, income status, insurance status) were matched using a 1:1 nearest neighbor propensity score with 0.05 caliper width (Appendix A) [24,25,26]. On matched cohort, a secondary multivariate regression model was built as described above. All analysis was performed using Stata software version 17.0 (Stata Corporation, College Station, TX, USA). *p* values of less than 0.05 were considered statistically significant.

## 3. Results

### 3.1. Demographics and Baseline Comorbidities

From January 1, 2020, to December 31, 2020, the N.I.S. included 1,659,040 hospitalizations with a COVID-19 infection diagnosis listed during an inpatient episode of care. Of these patients, 4501 (0.27%) were positive for Influenza co-infection, and 1,654,539 (99.73%) were influenza-negative. Patient characteristics, including demographics and comorbidities, are summarized in Table 1.

Prior to propensity matching, patients who were both COVID and Influenza positive tended to be older with an age of more than 50 years (82.9% for Influenza positive and vs. 78.2% for Influenza negative cohort, *p* < 0.001), have lower household income (41.9% for Influenza positive cohort had a median household income of less than $50,000 vs. 34.1% for Influenza negative cohort, *p* < 0.001), private insurance (32.2% vs. 27.6%, *p* < 0.001), were more likely to be from South Atlantic (30.6% vs. 20%, *p* < 0.001), East South Central (10.3% vs. 6.7%, *p* < 0.001) or West South Central (22.8% vs. 14.3%, *p* < 0.001) regional hospitals, needed a small bed-size (30.1% vs. 24.3%, *p* = 0.002) and were treated at rural hospitals (16% vs. 9.8%, *p* < 0.001) relative to the COVID positive and Influenza negative cohort. The influenza-positive cohort was also more likely to suffer from uncomplicated hypertension and acute kidney injury, as determined by ICD-10 codes. No statistically significant difference was observed between the 2 cohorts for gender distribution (*p* = 0.19), race distribution (*p* = 0.49), comorbidities including coronary artery disease (*p* = 0.57), congestive heart failure (*p* = 0.58), diabetes mellitus uncomplicated (*p* = 0.87) or complicated (*p* = 0.29), chronic pulmonary disease (*p* = 0.62), obesity (p = 0.98), smoking (*p* = 0.22), stroke (*p* = 0.76), and cardiac arrest (*p* = 0.25).

### 3.2. In-Hospital Outcomes

The odds of needing mechanical ventilation (21.1% for Influenza positive and 15.9% for influenza negative, OR 1.42, 95% CI 1.19–1.69, *p* < 0.001), higher in-hospital mortality (15.7% for Influenza positive and 13.4% for Influenza negative with OR 1.29, 95% CI 1.05–1.58, *p* = 0.01), mean hospitalization cost ($128,617 for Influenza positive and $91,735 for Influenza negative with *p* < 0.001), and total length of stay (9.8 days for Influenza positive and eight days for Influenza negative with *p* < 0.001), were statistically significantly higher for the influenza-positive cohort. No statistically significant difference was observed between the 2 groups for disposition post-discharge from the hospital (*p* = 0.44) and use of vasopressor (*p* = 0.83) (Table 2).

Further breakdown of in-hospital mortality prior to propensity matching showed higher in-hospital mortality for patients in Influenza positive cohort admitted to rural hospitals (12.8% vs. 7.8%, *p* = 0.03). No statistically significant difference was observed between the two cohorts for in-hospital mortality in regard to gender (*p* = 0.65), age (*p* = 0.7 to 1), or racial distribution (*p* = 0.07 to 0.93) (Table 3).

### 3.3. Propensity-Matched Demographics and Baseline Comorbidities

All covariates from Table 1 and Table 2 were used to generate a propensity score. We matched 4085 COVID and influenza-positive patients with 4085 COVID-positive and Influenza negative patients. The matched cohorts were assessed for covariate balance. As shown in Table 4, propensity matching eliminated almost all significant differences between the two groups in regard to clinical characteristics, demographics and baseline comorbidities except for hospital division and setting. Patients seen in influenza-negative (and COVID-positive) were more likely to be from New England (75.4% vs. 12.2%, *p* < 0.001) in mid-Atlantic (21.9% vs. 9.0%, *p* < 0.001) regional hospitals. Patients in Influenza positive cohort were more likely to be treated at rural hospitals (15.9% vs. 3.7%, *p* < 0.001) relative to the COVID-positive and Influenza negative cohort patients.

### 3.4. Propensity-Matched in-Hospital Outcomes

In propensity-matched analysis, similar in-hospital outcomes were observed when compared to unmatched data. That is, a Statistically significant difference was observed in regard to the odds of needing mechanical ventilation (20.9% for Influenza positive and 15.4% for influenza negative, OR 2.01, 95% CI 1.19–3.39, *p* = 0.009), higher in-hospital mortality (16.3% for Influenza positive and 14.6% for Influenza negative with OR 2.09, 95% CI 1.03–4.24, *p* = 0.04), mean hospitalization cost ($129,742 for Influenza positive and $68,878 for Influenza negative with *p* = 0.04), and total length of stay (9.9 days for Influenza positive and 8.2 days for Influenza negative with *p* = 0.01), between the two patient cohorts. In addition, a statistically significant difference was observed between the two groups for disposition post-discharge from the hospital (62.8% of Influenza positive patients discharged home vs. 52.2% of Influenza negative patients, *p* = 0.004) (Figure 1). Similar to unmatched data, no statistically significant difference was observed between the two groups for the use of vasopressor (2.3% vs. 3.9%, *p* = 0.18) in propensity-matched data (Table 5).

## 4. Discussion

Between 1 January to 31 December 2020, 165,9040 patients were identified with a diagnosis of COVID-19. This study showed that patients with a co-diagnosis of COVID-19 and Influenza had a significantly higher mean hospitalization cost ($129,742 vs. $68,878, *p* = 0.04) and total length of stay (9.9 days vs. 8.2 days, *p* = 0.01), higher odds of needing mechanical ventilation (OR 2.01, 95% CI 1.19–3.39), and overall higher in-hospital mortality (OR 2.09, 95% CI 1.03–4.24) relative to the COVID positive and Influenza negative cohort. To the best of our knowledge, this is the largest sample of hospitalized patients evaluated for COVID-19 and Influenza coinfection [27,28,29]. Various prior studies, often limited to case reports, case series, or review articles based on the pooling of multiple studies with data obtained using disparate methods, may limit the clinical utility and generalizability of those results.

Of 1,659,040 patients identified with COVID-19 between 1 January 2020, to 31 December 2020, only 4501 (0.27%) of patients were identified to have a co-diagnosis of Influenza. Similarly, a low rate ranging from 0.54% to 0.8% has been reported in the literature [12,30]. Some authors have suggested that this low rate of coinfection may, in part, be due to the impact of the COVID-19 pandemic on the circulation of certain respiratory viruses, including nonpharmaceutical interventions like facemasks, stay-at-home orders, closure of schools, and local-national borders, hand hygiene, and environmental cleaning [31,32,33].

It is crucial to remember that SARS-CoV-2 and influenza viruses share similar transmission dynamics as well as have overlapping clinical features, including fever, muscle aches, and dry cough. Therefore, it can be difficult to distinguish between early-stage COVID-19 and influenza infection based on symptomatology alone [34,35,36]. However, the treatment of these two infections is different. To make matters even more complicated, a small proportion of patients can present with coinfection with COVID-19 and Influenza. Therefore, early detection and accurate distinction between the two infections are paramount for appropriate management. To address this need, various testing kits have been developed with the ability for qualitative detection and differentiation of SARS-CoV- 2, Influenza A & B, and respiratory syncytial virus (R.S.V.) such as Xpert Xpress SARSCoV- 2/Flu/R.S.V. (Xpert 4-in1) assay; Xpert Xpress S.A.R. S-CoV-2/Flu/R.S.V.; PowerChek SARS-CoV-2, and Influenza A&B, R.S.V. Multiplex Real-time PCR Kit [37,38,39,40].

Although various studies with relatively small sample sizes have reported poor clinical outcomes in patients with influenza virus and SARS-CoV-2 coinfection, the exact extent of its clinical impact remains ill-explored [13,14,41]. In our analysis, a significantly higher in-hospital mortality (OR 2.09, 95% CI 1.03–4.24, *p* = 0.01) was seen in the influenza virus and SARS-CoV-2 coinfected cohort relative to the COVID-positive and Influenza negative cohort. The compounding effect of SARS-CoV-2 and influenza virus infection could be, in part, explained by the fact that both viruses primarily impact the alveolar type II cells (AT2 pneumocytes). Therefore, a coinfection with these viruses can exacerbate the potential respiratory epithelial damage [41,42]. This observation is supported by findings in our study, which showed patients with COVID-19 and Influenza coinfection were more likely to need mechanical ventilation (OR 2.01, 95% CI 1.19–3.39). Corroborative findings were noted in a similar UK-based study on SARS-CoV-2 infection co-infection with influenza viruses, respiratory syncytial virus, or adenoviruses in 212 466 adults who were admitted to hospital in the UK between 6 February 2020, and 8 December 2021, the authors found Viral co-infection was detected in 583 (8·4%) patients: 227 patients had influenza viruses, 220 patients had the respiratory syncytial virus, and 136 patients had adenoviruses. They noted that coinfection with influenza viruses was associated with increased odds of receiving invasive mechanical ventilation and mortality compared with SARS-CoV-2 mono-infection [43].

In a meta-analysis including over 3070 patients with the diagnosis of COVID-19. The authors noted that 76 patients (0.8%) were diagnosed coinfected with influenza [12]. In another review of the literature, including over 1103 patients with the diagnosis of COVID-19, only six patients (0.54%) were noted to have an influenza coinfection. Neither of these studies reported on the outcomes and complications, including mortality, in-hospital complications, length of stay, and healthcare utilization costs [30]. We found that the COVID-19 and Influenza coinfection cohort had an average of $60,864 more in-hospital charges ($129,742 vs. $68,878 for the negative influenza cohort). Although these hospital charges are not directly related to the patient costs, however, they still give a strong indication of the overall healthcare financial burden. Factors that may have contributed to higher hospital charges may include the longer length of stay (9.9 days vs. 8.2 days, *p* = 0.01) and complex clinical course leading to increased use of hospital resources.

Interestingly, in our study, we didn’t notice any statistically significant difference between gender distribution (*p* = 0.19), race distribution (*p* = 0.49), and comorbidities, including coronary artery disease (*p* = 0.57), congestive heart failure (*p* = 0.58), diabetes mellitus uncomplicated (*p* = 0.87) or complicated (*p* = 0.29), chronic pulmonary disease (*p* = 0.62), obesity (*p* = 0.98), smoking (*p* = 0.22), stroke (*p* = 0.76), and cardiac arrest (*p* = 0.25). An incidental significance was seen between the two cohorts regarding the presence of uncomplicated hypertension and acute kidney injury. However, the underlying clinical relevance of this, if any, remains unclear.

### 4.1. Vaccination and Vaccine Hesitancy

With the ongoing COVID-19 pandemic and upcoming seasonal influenza epidemics. Coinfections of these respiratory viruses remain a significant concern from an individual and public healthcare perspective, especially in light of poor patient outcomes and increased healthcare costs [13,14]. Therefore, early prophylactic measures, including vaccination against COVID-19 and Influenza, and nonpharmaceutical interventions (NPIs) like physical distancing and face masks may help reduce disease burden and improve clinical outcomes.

Since December 2020 (the data used for this study), multiple safe and effective COVID-19 vaccines have become available for public use. Over 68% of the world population has received at least one dose of a COVID-19 vaccine, with a total of over 12 billion doses have been administered globally [14,44]. With this successful administration of the COVID-19 vaccine, public life is slowly returning to the pre-COVID-19 baseline. Further discussion on COVID-19 vaccination remains beyond the purview of this article.

Influenza vaccination has also been shown to be associated with a lower risk of SARS-CoV-2 infection [45,46,47]. For instance, in one meta-analysis, including 16 studies with 290,327 participants, Influenza vaccination was associated with a lower risk of SARS-CoV-2 infection (pooled adjusted OR: 0.86, 95%CI: 0.81–0.91) [48]. In another retrospective U.K.-based study, with 6921 COVID-19 positive (between 1 January–31 and July 2020) participants, a 15–24% lower odds of hospitalization and all-cause mortality was seen in 2613 (38%) participants who received influenza vaccine prior to diagnosis of COVID-19. The author postulated that this direct protective effect against COVID-19 from influenza vaccination might, in part, be explained by vaccination’s role in inducing adaptive/innate immunity against a wider group of unrelated pathogens, similar to the protective immune response following B.C.G. and measles vaccines [49,50,51,52,53,54]. Therefore, the Influenza vaccine is recommended by various national and international health authorities, including World Health Organization (WHO), in particular among high-risk groups, including healthcare workers, to help minimize the influenza burden as well as allow better preparedness for COVID-19 waves [36,55].

It is essential to highlight that despite the availability of safe and effective vaccines against COVID-19 and Influenza, further work needs to be done to address and improve vaccine hesitancy. Various steps, including rebuilding public vaccine confidence by uniform, consistent, transparent, and effective communication from policymakers, media, and health care providers, and effective community engagement are essential to improve vaccine acceptance [56,57,58].

Fortunately, studies have noted a shift in influenza vaccine acceptance globally since the beginning of the COVID-19 pandemic. In a meta-analysis, including 27 studies with 39,193 participants, an increase in intention to vaccinate (1.50, 95% CI 1.32–1.69, *p* < 0.001) in season 2020/2021, regardless of age, gender, and occupation. The author noted that the COVID-19 pandemic had increased the intention to vaccinate against Influenza internationally and may provide a new window of opportunity to further promote influenza vaccination and decrease vaccine hesitancy [59,60,61,62,63,64].

Various studies have also highlighted that public health interventions imposed during the COVID-19 pandemic to curb the rapidly spreading SARS-CoV-2 virus with measures including physical distancing, facemasks, stay-at-home orders, closure of schools and local-national borders, hand hygiene, and environmental cleaning may help prevent transmission of R.S.V. and influenza virus as well [31,32]. However, post the advent and successful administration of COVID-19 vaccination across the globe, countries began to relax restrictions imposed during the COVID-19 pandemic. This, in turn, will also increase exposure to various infectious viruses, including Influenza and R.S.V. as well. Based on epidemiological predictive models, some authors have predicted that more severe influenza and R.S.V. epidemics with higher rates and more severe clinical course of Influenza should be expected in season 2021–2022, following low activity in prior and population exposure in season 2020–2021 [28,29,65,66,67]. This phenomenon is also known as Immunity debt, defined as a lack of immune stimulation and protective immunity due to reduced exposure to a given pathogen, leaving a greater proportion of the population susceptible to the given pathogen in the future [31,68,69,70,71,72]. These findings further strengthen the argument for public health campaigns and policy initiatives with particular emphasis on mass immunization.

### 4.2. Limitations

There are several limitations to our study. There are inherent limitations due to the retrospective nature of the study. In addition, the database used for this study, the N.I.S., is an administrative database and, therefore, may not fully represent the complex nature of the patient’s hospital course, including information on history, physical exam, lab values, and imaging/pathological investigation reports. These inpatient results may not be representative of the outpatient trends. It is also important to highlight that these results were calculated prior to the advent and administration of successful COVID-19 vaccines and, therefore, may not represent current hospitalization trends.

## 5. Conclusions

Using a large, nationally representative sample of patient data comparing COVID-19 and Influenza positive with COVID-19-positive and Influenza negative cohorts, we found that Influenza negative cohort had lower overall mortality, shorter hospital stays, and lower hospital charges. Our findings Our study emphasizes the importance of screening for co-infecting viruses in COVID-19 patients to allow early intervention and management with appropriate antivirals. It also highlights the importance of COVID-19 and Influenza vaccination in preventing reduce the risk of severe outcomes and mortality in the ongoing COVID-19 pandemic.

## Figures and Tables

**Figure 1 vaccines-10-02159-f001:**
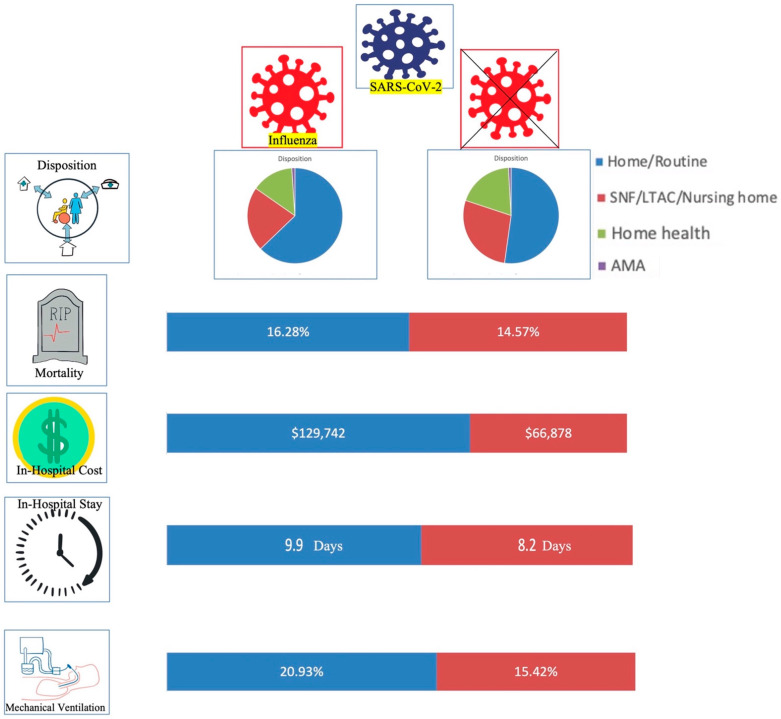
Influenza and COVID-19 patient in-hospital outcomes. SNF = skilled nursing facility, LTAC = Long-term acute care, AMA = against medical advice.

**Table 1 vaccines-10-02159-t001:** Influenza and COVID-19 unmatched patient-level characteristics. SD = Standard deviation, CAD = Coronary Artery disease, CHF = Congestive heart failure, HTN = Hypertension, DM = Diabetes Mellitus, AKI – Acute kidney injury, HD = Hemodialysis.

CHARACTERISTICS	Influenza Positive	Influenza Negative	*p* Value
N = 165,9040	N = 4501 (0.27%)	N = 165,4539 (99.73%)	
SEX (Female)	45.67%	47.95%	0.19
Mean age years (SD)			0.2
Male	62.88 (15.33)	63.42 (16.2)	
Female	65.2 (16.1)	63.06 (18.8)	
age groups			<0.001
≥18–29	2.67%	4.95%	
30–49	14.44%	16.81%	
50–69	42.33%	37.22%	
≥70	40.56%	41.02%	
RACE			0.49
Caucasians	47.72%	50.92%	
African American	21.64%	19.05%	
Hispanics	21.53%	21.47%	
Asian or Pacific Islander	3.19%	3.25%	
Native American	1.03%	1.03%	
Others	4.9%	4.28%	
median household income			<0.001
<49,999$	41.89%	34.1%	
50,000–64,999$	26.8%	27.2%	
65,000–85,999$	18.02%	22.17%	
>86,000$	13.29%	16.53%	
insurance status			0.003
Medicare	52.54%	53.24%	
Medicaid	11.45%	15.17%	
Private	32.23%	27.63%	
Self-pay	3.78%	3.96%	
hospital division			<0.001
New England	1.33%	3.8%	
Middle Atlantic	8.67%	14.63%	
East North Central	7.78%	15.56%	
West North Central	4.22%	6.75%	
South Atlantic	30.56%	20.05%	
East South Central	10.33%	6.7%	
West South Central	22.78%	14.3%	
Mountain	5.22%	6.92%	
Pacific	9.11%	11.3%	
hospital bedsize			0.002
Small	30.11%	24.31%	
Medium	31.11%	28.98%	
Large	38.78%	46.7%	
hospital teaching status			<0.001
Rural	16%	9.79%	
Urban non-teaching	21%	18.65%	
Urban teaching	63%	71.56%	
comorbidities			
CAD	18.67%	17.96%	0.57
CHF	16.89%	17.57%	0.58
HTN uncomplicated	46.33%	38.11%	<0.001
HTN complicated	24%	27.07%	0.03
DM uncomplicated	14.56%	14.74%	0.87
DM complicated	24.89%	26.36%	0.29
Chronic pulmonary disease	22.67%	22%	0.62
Obesity	25.67%	25.64%	0.98
Smoking	23.78%	25.61%	0.22
AKI	32.78%	28.63%	0.007
AKI requiring HD	3%	2.44%	0.31
Stroke	1.44%	1.57%	0.76
Cardiac arrest	3.44%	2.79%	0.25

**Table 2 vaccines-10-02159-t002:** In-hospital outcomes. SNF = skilled nursing facility, LTAC = Long-term acute care, AMA = against medical advice.

VARIABLE	Influenza Positive	Influenza Negative	*p* Value
Disposition			0.44
Home/Routine	63.61%	61.06%	
SNF/LTAC/Nursing home	21.48%	22.19%	
Home health	13.95%	15.4%	
AMA	0.96%	1.35%	
Vasopressor use	2.33%	2.65%	
	Adjusted odds ratio ^1^ 0.94 (95% CI 0.58–1.55)		0.83
Mechanical ventilation	21.11%	15.89%	
	Adjusted odds ratio ^1^ 1.42 (95% CI 1.19–1.69)		<0.001
In-hospital mortality (N = 222,490)	15.68%	13.41%	
	Adjusted odds ratio ^1^ 1.29 (95% CI 1.05–1.58)		0.01
Mean total hospitalization charge ($)	128,617$	91,735$	
	Adjusted total charge ^1^ 39,045 $ higher		<0.001
Mean length of stay (days)	9.8	8.03	
	Adjusted length of stay ^1^ 1.8 day higher		<0.001

^1^ Adjusted for age, sex, race, income level, insurance status, discharge quarter, elixhauser co-morbidities, hospital location, teaching status, and bed size.

**Table 3 vaccines-10-02159-t003:** Mortality breakdown (for before matching sample).

Variable	Influenza Positive	Influenza Negative	*p* Value
total died (222,490)	706	221,784	
sex			0.65
Male	56.74%	58.47%	
Female	43.26%	41.53%	
age groups			
≥18–29	0.71%	0.6%	0.86
30–49	5.67%	4.93%	0.7
50–69	30.5%	30.52%	0.99
≥70	63.12%	63.94%	0.84
race			
Caucasians	44.68%	51.51%	0.13
African American	19.15%	16.86%	0.47
Hispanics	26.24%	19.57%	0.07
Asian or Pacific Islander	2.13%	3.35%	0.42
Native American	2.13%	1.22%	0.32
Others	4.26%	4.4%	0.93
hospital teaching status			
Rural	12.77%	7.81%	0.03
Urban non-teaching	17.02%	17.74%	0.83
Urban teaching	70.21%	74.45%	0.28

**Table 4 vaccines-10-02159-t004:** Influenza and COVID-19 propensity 1:1 matched patient-level characteristics. SD = Standard Deviation.

CHARACTERISTICS	Influenza Positive	Influenza Negative	*p* Value
N = 8170	N = 4085	N = 4085	
SEX (Female)	47%	47.12%	0.95
Mean age years (SD)	64.63 (10.01)	64.67 (10.01)	0.96
age groups			0.99
≥18–29	2.57%	2.57%	
30–49	13.34%	13.22%	
50–69	41.25%	41.25%	
≥70	42.84%	42.96%	
race			0.99
Caucasians	47.98%	47.98%	
African American	21.18%	21.18%	
Hispanics	21.91%	21.91%	
Asian or Pacific Islander	3.18%	3.3%	
Native American	0.98%	0.73%	
Others	4.77%	4.9%	
median household income			0.99
<49,999$	42.47%	42.35%	
50,000–64,999$	26.81%	26.81%	
65,000–85,999$	17.75%	17.87%	
>86,000$	12.97%	12.97%	
insurance status			0.99
Medicare	53.37%	53.37%	
Medicaid	11.38%	11.38%	
Private	31.46%	31.58%	
Self-pay	3.76%	3.67%	
hospital division			<0.001
New England	12.2%	75.4%	
Middle Atlantic	9.06%	21.91%	
East North Central	7.96%	1.1%	
West North Central	4.16%	0.37%	
South Atlantic	30.48%	0.37%	
East South Central	10.53%	0.12%	
West South Central	22.64%	0.37%	
Mountain	4.9%	0.37%	
Pacific	9.06%	0%	
hospital bedsize			0.89
Small	29.62%	30.48%	
Medium	31.82%	32.93%	
Large	38.56%	36.6%	
hospital teaching status			<0.001
Rural	15.91%	3.67%	
Urban non-teaching	21.42%	13.34%	
Urban teaching	62.67%	82.99%	

**Table 5 vaccines-10-02159-t005:** In-hospital outcomes for propensity 1:1 matched sample. SNF = skilled nursing facility, LTAC = Long-term acute care, AMA = against medical advice.

VARIABLE	Influenza Positive	Influenza Negative	*p* Value
Disposition			0.004
Home/Routine	62.82%	52.21%	
SNF/LTAC/Nursing home	21.85%	27.88%	
Home health	14.26%	19.03%	
AMA	1.06%	0.88%	
Vasopressor use	2.33%	3.92%	
	Adjusted odds ratio ^1^ 2.12 (95% CI 0.7–6.42)		0.18
Mechanical ventilation	20.93%	15.42%	
	Adjusted odds ratio ^1^ 2.01 (95% CI 1.19–3.39)		0.009
In-hospital mortality (N = 1260)	16.28%	14.57%	
	Adjusted odds ratio ^1^ 2.09 (95% CI 1.03–4.24)		0.04
Mean total hospitalization charge ($)	129,742$	66878$	
	Adjusted total charge ^1^ 57,355 $ higher		0.04
Mean length of stay (days)	9.9	8.2	
	Adjusted length of stay ^1^ 2.7 day higher		0.01

^1^ Adjusted for discharge quarter, elixhauser co-morbidities, hospital location, teaching status, and bed size.

## Data Availability

Not applicable.

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
