# Peer review of "COVID-19 and Influenza Coinfection Outcomes among Hospitalized Patients in the United States: A Propensity Matched Analysis of National Inpatient Sample"

_vaccines, 2022, doi:10.3390/vaccines10122159_

Round 1

Reviewer 1 Report

The article presented by Garg et al. is interesting and shows a lot of work in collecting and analyzing data. However, it has two major methodological limitations that may bias the conclusions it presents.

1. The study is carried out on a very large number of patients. under these conditions a minor effect can be "seen" by a p-value and it is preferable to use 95% confidence intervals to determine whether an effect is real or not. Please modify

2. To compare the severity of outcomes in patients with COVID-19 and COVID-19/Influenza infection, it is absolutely necessary to take into account the risk factors of the patients. It is quite obvious that an immunocompromised and obese patient with COVID-19 is not comparable to a young and healthy patient with COVID-19/Influenza. In the same way, it is also absolutely necessary to take account of vaccination status. Please modify or discuss

Author Response

Dear reviewer, Thanks for your valuable suggestions. Please see our responses below. 

  1. The study is carried out on a very large number of patients. under these conditions a minor effect can be "seen" by a p-value and it is preferable to use 95% confidence intervals to determine whether an effect is real or not. Please modify

Thank you for your input. Due to the very large sample size, even minor differences among groups can be seen. However due to complexity of data set and survey analysis, STATA doesn’t normally provide confidence intervals when chi square test is implemented. We did represent 95% CI for all our outcomes.   

  1. To compare the severity of outcomes in patients with COVID-19 and COVID-19/Influenza infection, it is absolutely necessary to take into account the risk factors of the patients. It is quite obvious that an immunocompromised and obese patient with COVID-19 is not comparable to a young and healthy patient with COVID-19/Influenza. In the same way, it is also absolutely necessary to take account of vaccination status. Please modify or discuss

Thank you for your astute observation. We agree that a number of risk factors, including comorbidities, can influence the severity of outcomes in these patient cohorts. However, the database used for this analysis, that is the N.I.S., is an administrative database, and, therefore, it’s not feasible to collect data specific to certain risk factors. We have highlighted that as a limitation of our study. We have, however included information on comorbidities (table 1) and their distribution between these two cohorts. All covariates from Tables 1 and 2 were used to generate a propensity score. We matched 4085 COVID and influenza-positive patients with 4085 COVID-positive and Influenza negative patients. The matched cohorts were assessed for covariate balance. As shown in table 4, propensity matching eliminated almost all significant differences between the two groups in regard to clinical characteristics, demographics and baseline comorbidities except for hospital division and setting. Further, we ran a multivariate regression analysis on the matched cohort to adjust for confounders (Elixhauser comorbidities, hospital characteristics) Table 2. This as reported in the footnotes under Table 2.  Further Elixhauser comorbidities include obesity, Hypertension, and Diabetes, among many others.

Regarding the vaccination status. As the data used in this study only represent patients admitted to US hospitals in 2020. That is prior to the advent and administration of successful COVID-19 vaccines and, therefore, may not represent current hospitalization trends. We have also highlighted this as a limitation of our study.

At present, NIS data is only available from year 2020, which was released in October 20. We hope in the future, additional studies can build upon these results and study the impact of COVID-19 vaccination on patient outcomes, among other questions.

Sincerely 

RS 

Reviewer 2 Report

The manuscript entitled " COVID-19 and Influenza Coinfection Outcomes Among Hospitalized Patients in the United States: A Propensity Matched Analysis of National Inpatient Sample" describes the disease outcome in influenza and covid-19 patients. A lot of similar studies have been carried out. The study also lacks novelty. The authors have included "Vaccination and Vaccine Hesitancy" in the manuscript, however, i feel this is not matching their objective of study. 

Author Response

Dear reviewer, thanks for your valuable suggestions. 

Suggestion 1. The manuscript entitled " COVID-19 and Influenza Coinfection Outcomes Among Hospitalized Patients in the United States: A Propensity Matched Analysis of National Inpatient Sample" describes the disease outcome in influenza and covid-19 patients. A lot of similar studies have been carried out. The study also lacks novelty.

Dear reviewer, thank you for your time and effort in helping us improve this paper. We agree that multiple studies have tried to address a similar questions. However, we believe this is the first study done on a national database including over 1,659,000 hospitalizations for all geographic locations and including a diverse patient population, we believe it is a true representative sample from USA. The previously published data on coinfection is some studies had n of 583 - 4 patients and we studied 4085 patients and to our knowledge, this is the largest study of this kind.   

583 - SARS-CoV-2 co-infection with influenza viruses, respiratory syncytial virus, or adenoviruses - The Lancet

176 - The epidemiology and clinical characteristics of co-infection of SARS-CoV-2 and influenza viruses in patients during COVID-19 outbreak - PubMed (nih.gov)   202 - Frontiers | COVID-19 and Influenza Co-infection: A Systematic Review and Meta-Analysis (frontiersin.org)   34- Influenza co-infection associated with severity and mortality in COVID-19 patients | Virology Journal | Full Text (biomedcentral.com)   13- Are coinfections with COVID‐19 and influenza low or underreported? An observational study examining current published literature including three new unpublished cases - Antony - 2020 - Journal of Medical Virology - Wiley Online Library   4  patient SARS-CoV-2 and influenza virus co-infection (nih.gov)   97- Co‐infection of influenza A virus and SARS‐CoV‐2: A retrospective cohort study - Cheng - 2021 - Journal of Medical Virology - Wiley Online Library    

Given such large data sample, we believe we have presented a more robust comparative analysis of clinical features and outcomes among U.S. adults admitted to the hospital with COVID-19 and influenza infection.

The authors have included "Vaccination and Vaccine Hesitancy" in the manuscript, however, i feel this is not matching their objective of study. 

Thank you for your astute observation. We have modified the study objectives to reflect those changes. “In this study, we aim to utilize the United States National Inpatient Sample (N.I.S.) database from 2020 to provide more robust comparative data on clinical features and outcomes among U.S. adults admitted to the hospitalist with COVID-19 and influenza infection as well as discuss the reasons for vaccine hesitancy and its potential impact on healthcare policies.”

We also modified the conclusion as suggested by the reviewer.  Please see the conclusion below 

Using a large, nationally representative sample of patient data comparing COVID-19 and Influenza positive with COVID-19-positive and Influenza negative cohorts, we found that Influenza negative cohort had lower overall mortality, shorter hospital stays, and lower hospital charges. Our findings Our study emphasizes the importance of screening for co-infecting viruses in COVID-19 patients to allow early intervention and management with appropriate antivirals. Given poor outcomes in patients with influenza and COVID-19 coinfection, it is essential to highlight the need for continued efforts towards boosting herd immunity by ongoing vaccination against both infections

sincerely 

RS 

Round 2

Reviewer 1 Report

All comments have been adressed

Reviewer 2 Report

The authors have modified the manuscript and it may be accepted in the present format.